# Development of a Technology for Protein-Based, Glueless Belevskaya Pastille with Study of the Impact of Probiotic Sourdough Dosage and Technological Parameters on Its Rheological Properties

**DOI:** 10.3390/foods12193700

**Published:** 2023-10-09

**Authors:** Yuliya Pronina, Talgat Kulazhanov, Zhanar Nabiyeva, Olga Belozertseva, Anastasiya Burlyayeva, Alberto Cepeda, Erik Askarbekov, Gulzhan Urazbekova, Elmira Bazylkhanova

**Affiliations:** 1Department of Information and Patent Research, Almaty Technological University, Almaty 050000, Kazakhstan; medvezhonok_87@inbox.ru (Y.P.); tkulazhanov_atu@mail.ru (T.K.); 66bel@bk.ru (E.B.); 2Food Safety Research Institute, Almaty Technological University, Almaty 050000, Kazakhstan; atu_nabiyeva@mail.ru (Z.N.); nik_belozerceva@mail.ru (O.B.); n.burlyaeva29@gmail.com (A.B.); gule90@mail.ru (G.U.); 3Laboratorio de Higiene Inspección y Control de Alimentos, Departamento de Química Analítica, Nutrición y Bromatología, Universidad de Santiago de Compostela, 27002 Lugo, Spain; 4Department of Technology of Bread Products and Processing Industries, Almaty Technological University, Almaty 050000, Kazakhstan; erik_ab82@mail.ru; 5Department of Food Technology, Almaty Technological University, Almaty 050000, Kazakhstan

**Keywords:** *Lactobacillus acidophilus*, moisture content, drying time, penetration rate, confectionery

## Abstract

The proper functioning of the gastrointestinal tract plays an important role in strengthening the immune system. It is an undeniable fact that lactic acid microorganisms are necessary for the proper functioning of the gastrointestinal tract, the source of which are mainly dairy products. However, there is a problem with the digestibility of lactose; therefore, alternative sources and carriers of probiotics are of particular interest. Due to its dietary and natural properties, protein marshmallow can serve as such a carrier. Therefore, the direction of this study is to identify the dependence of technological factors on the rheological properties of the product and the growth of lactic acid microorganisms in confectionery products enriched with lyophilised strains. According to the results of the study, the following was determined: the optimal technology to produce enriched Belevskaya pastille with a mixture of *Lactobacillus acidophilus* makes it possible to obtain a product with the necessary rheological properties, utilising a mass drying mode in a dehydrator at 50 °C for 16 h. The strains *L. acidophilus* M3 and *L. acidophilus* M4 were the most resistant to a high concentration of bile (40%) in the substrate. Based on the analysis of variance and the obtained regression equations, it was revealed that the growth of lactic acid microorganisms in the product was strongly influenced by the amount of ferment introduced (R² = 0.96). The level of penetration is influenced by factors such as the amount of probiotic starter introduced, the drying time and the interaction of drying time factors on the amount of starter added. The higher the level of penetration, the crumblier the product. The resulting functional product can be characterized as symbiotic since the main raw material of plant origin contains a large amount of fibre, which acts as a prebiotic, and the strain of microorganism, which acts as a probiotic. The data described in the article can be applied in the technological processes of similar products to regulate the structure of the product and vary the dosage of enrichment with probiotic starter cultures.

## 1. Introduction

Probiotics have been reported in recent decades to have large benefits for human health, and consequently, functional products based on probiotics are becoming extremely popular due to their unique properties that positively affect the body [1]. Dairy products are the main product for enrichment with probiotics [1]. However, many individuals are lactose intolerant [2], and hence, the fortification of non-dairy-based products needs to be expanded. Since confectionery products are in demand among the population, this segment should be considered as a source of enrichment with probiotics.

Belevskaya whipped protein pastille, due to its dietary and natural properties, can be an alternative carrier for enrichment with probiotic cultures. Belevskaya whipped protein pastilles are confectionery obtained by churning fruit and berry puree with sugar and egg whites [1] and then mixing with a hot glutinous syrup of sugar, molasses and agar or marmalade mass; both pastille and marshmallow belong to this type of product [1,2].

For a long time, the production of whipped protein pastilles has been considered one of the ways to preserve fruit and berry raw materials [3]. It has also been noted that in the original version of the recipe, there were no sweeteners; only later, to increase the sweetness, honey was added to the pastille mass, which was then replaced with sugar [3].

The main raw material for the production of whipped protein pastilles is apple puree, which is obtained from baked apples, in which a special form of fibre is pectin. The action of pectin is more pronounced during the heat treatment of apples; this process is accompanied by the acquisition of a gel-like sheen on the surface of apples. It is known that the use of applesauce as the main prescription component carries a number of positive aspects. Emerging evidence suggests that pectin may help repair and preserve the intestinal mucosa [4]. Pectin also helps in the modulation of intestinal bacteria, eliminates toxins in the intestines and reduces inflammation [5].

A study was conducted showing that applesauce from 12 varieties of apples, manufactured industrially (heat treatment at 95 °C, 2 min, pasteurization at 90 °C, 5 min, cooling at 20 °C, 20 min), in its chemical and nutritional properties is very close to fresh apple pulp [6].

The topic of probiotic products is an integral part of the nutrition industry. Eating confectionery products produced using lactic acid bacteria (LAB) contributes to positive changes in the intestinal microbiota.

Previous research [7] showed that, despite some loss of probiotic adhesion, the combined presence of extract and probiotic is more effective in reducing the overall amount of adhered viable pathogen cells than the probiotic alone, regardless of the probiotic/pathogen system considered.

A review article [8] shows that probiotics hold the potential to increase innate and acquired immunity and activate anti-inflammatory and anti-allergic effects. This study shows the relevance of the use of probiotics in human life. However, the question remains unresolved regarding how easy and affordable it is to consume them in everyday life, in particular for people suffering from the indigestibility of milk protein.

In [9], the authors proved the possibility of producing probiotic grape marmalade without the use of dairy products.

The authors [10] studied the influence of technological parameters in the production of marmalade products on the safety of vital nutrients. In their further works [11], the morphology of the probiotic culture was studied and it was found that the volume of whey (250 mL) affects the growth of lactic acid microorganisms in the marmalade product. The optimal amount of probiotic culture to be added to the serum was 0.01–0.02 g, while the time of reviving the microorganisms was 6 h. As a result of the study, the confectionery product contained from 1 to 3 CFU/g of lactic acid culture, and 1.7 and 2.2 times more antioxidants compared to the control, which justifies the benefits for the gastrointestinal tract and maintaining human immunity, due to the production of its own interferon.

The studies conducted in this area prove the promise of this direction; however, before industrial applicability, it is necessary to study technological methods for introducing a probiotic culture as an active enriching ingredient, followed by a study of the safety of lactic acid bacteria (LAB) after technological processing and storage. It is also necessary to analyse exactly what effect the dosage and type of probiotic culture has on the rheological properties of finished products [12].

In order to identify the dependence of technological factors on the rheological properties of the product and the growth of lactic acid microorganisms in battered confectionery products, we set three objectives:

It is necessary to determine the optimal dosage of probiotic starter, which would allow us to enrich the product with probiotics and, at the same time, would not worsen the rheological and organoleptic properties of the finished products;The number of surviving LAB in bunched pastilles after processing treatment should be investigated to determine the effectiveness of their probiotic properties and stability in acidic medium;The influence of technological factors on the rheological properties of Belevskaya whipped protein pastille and the growth and development of LAB should be substantiated using mathematical modelling, revealing the significance of factors by means of dispersion and regression analysis.

## 2. Materials and Methods

### 2.1. Belevskaya Whipped Protein Pastilles Added with Probiotic Elaboration

A whipped confectionery product was made according to the standard [3], using whipped protein glueless pastille without sugar as the main raw material, which is applesauce from baked apples [3]. Such an almost equivalent substitute for egg white is a mixture of albumin, pasteurized egg white with increased whipping and a certain quantitative ratio of water.

According to the all-union collection of standards for Belevskaya (glueless) pastille, the addition of syrup with agar is not required and is interpreted in the definition by the words “or without adding them” [3,12]. One of the solutions to the issue of functional sweets is the complete exclusion of sugar from the composition of whipped protein glueless pastilles and the use of only baked apple puree with a high content of its own sugars and pectin, which, in combination with albumin of a high degree of churning, will be sufficient for structure formation [13]. Technology has been developed for a whipped protein pastille enriched with the Maxilin^®^ probiotic with a rational dosage between 0.009 and 0.021%. Strains were introduced in various mass fractions with probiotics in the free state, with immobilized probiotics introduced, and with pasteurized serum enriched with bacterial concentrate containing 0.001% acidophilus bacillus. The introduction of a probiotic culture was carried out at the stage of swollen pectin in a cooled syrup at a temperature below 45 °C. The technological factor of the route of introduction of probiotic culture at temperatures below 45 °C is of interest. In the known methods of marmalade production, at a temperature of 45 °C, the formation of jelly occurs, the rheological state of which will not allow the additive to be evenly distributed.

As part of the study, to identify the influence of technological factors on the rheological properties and the content of preserved lactic acid microorganisms in whipped protein pastilles, the following prototypes were developed (Table 1):

These samples were sent to determine the rheological properties—the level of penetration and moisture content—and to identify the remaining number of lactic acid microorganisms after technological processing.

### 2.2. Microbiological Determinations

Determination of the total number of mesophilic aerobic and facultative anaerobic microorganisms was carried out according to [14] on Nutrient Agar TM341 (Titan Biotech, Mumbai, India) and incubated at 30 °C for 72 h [15]. Determinations were carried out based on the accredited testing laboratory “Food Safety” of the Almaty Technological University on 14 samples of whipped confectionery products.

For the isolation of LAB, the sequential dilution method on agar was used. Ten grams of each sample was dissolved in 90 mL of MRS broth (Titan Biotech, Bhiwadi, India). After dissolving in MRS broth, the samples were shaken homogeneously and incubated at 37 °C for 24 h in an aerobic state. In plate serial agar, 10 mL of stock solution was suspended and mixed into 90 mL aqueous blanks to form a microbial suspension. Serial dilutions of 10^2^–10^6^ were made by pipetting 10 mL into 90 mL of aqueous blanks. Ten ml of each dilution was inoculated onto MRS agar plates (prepared by pouring 15 mL of sterile and chilled molten medium into plates) and incubated at 37 °C for 24 h for bacterial growth [16].

The heat resistance of LAB was determined by seeding 1% on skim milk in an amount of 0.1 mL per 10 mL of milk. Then, the test tube was placed in a water bath at a temperature of 60 °C and kept for 30, 60 and 90 min, after which it was rapidly cooled and subjected to thermostatting at a temperature of 37 °C for 48 h [17].

The resistance of LAB to bile was determined in a medium with different concentrations of bile salts (20%, 30% and 40%), and isolated bacterial cultures were incubated at 37 °C for 48 h. Then, 0.1 mL of the inoculum was transferred to MRS agar (Titan Biotech, Mumbai, India) in petri dishes and incubated at 37 °C for 48 h. The growth of LAB on MRS agar plates was used to identify isolates as resistant to bile salts [18].

To determine the pH resistance of LAB, samples were inoculated into sterile tubes with MRS broth (Titan Biotech, Mumbai, India) at different pH values (4.5–6.5) and incubated at 37 °C for 48 h. Then, 0.1 mL of the inoculum from each tube was transferred to MRS agar medium and incubated at 37 °C for 48 h [19].

Determination of the resistance of LAB to NaCl was carried out by seeding the cultures in liquid nutrient medium MRS with different contents of NaCl (2.0–6.0%), in which the bacteria were cultivated at 37 °C [20,21]. The morphological and cultural properties of LAB were determined using an electron microscope trinocular (halogen lamp) MS 300 according to conventional methods (micros HgmbH, Breitenfurter Strasse 386 A-1120, Vienna, Austria).

The classification of LAB into different genera was based on morphology and the way carbohydrates are fermented. Glucose, mannose, fructose, arabinose, maltose, sucrose, lactose, galactose, sorbitol, raffinose, xylose, mannitol, ribose and kramal were used as substrates (sugars and polyhydric alcohols). Microorganisms were seeded in a liquid medium with a 1% substrate and bromocresol purple at a concentration of 0.03 mg/mL (HiMedia, Mumbai, India). Analysis of the results of the bacterial enzymatic activity test was carried out by changing the colour (from violet to yellow) of the nutrient medium used [22,23,24]. When carrying out the identification, the determinant of the bacteria Bergey was used [25].

The acid-forming activity of lactic acid bacteria was determined by examining the acid-forming capacity and limiting acid formation. The acid-forming capacity of the cultures under study was determined by the rate of lactic acid formation per unit time (1 h). To determine the acid formation activity, isolates were seeded in 0.3 mL tubes with 10 mL of milk. The tubes were placed in a thermostat at 37 °C, after 24 h acidity was determined by a titrimetric method. For this purpose, 10 mL of the sample was diluted with 20 mL of distilled water and 1–2 drops of phenolphthalein indicator were added. Titration was carried out with 0.1 n NaOH solution until the appearance of a stable pink colour. Limit acid formation was determined by sowing 0.3 mL of the tested culture in 10 mL of skimmed milk. The cultures were thermostatted at 37 °C. After 7 days, the titratable acidity expressed in Turner degrees was determined [26].

### 2.3. Methods of Mathematical Processing of the Results of Experimental Studies

The planning of the experiment and the processing of experimental data were carried out using the application programs STATISTICA 13 (TIBCO Software, Palo Alto, CA, USA) and Microsoft Excel 2019 (Microsoft, Lemoyne Township, PA, USA).

In the mathematical processing of the analyses of the research experiment, a full-factorial standard plan was used: 2**(3-0), with the number of replicas equal to 2, resolution R = FULL, using 3 independent variables (factors), with 9 standard experiments, with a total number of experiments equal to 27 with an emphasis on 3 central points.

### 2.4. Method of Determination of Penetration

Hardness analysis (penetration force, N) was determined on an APN-360MG4 penetrometer designed to determine the depth of needle penetration (penetration) into the test sample at a given load and temperature. The following parameters were applied: penetration time (needle immersion), τ = 5 s; ambient air temperature, t = 25 °C; limits of permissible absolute error in measuring the movement of the needle ± 0.1 mm; and total weight (mass of the plunger with the needle and weight of the load), P = 100 g. The range of indications was from 0 to 400 penetration units. The depth of penetration of the needle (penetration) was determined by multiplying the obtained value of H by 10, rounded to the nearest whole number [27].

### 2.5. Moisture Determination Method

The moisture content of the samples was determined in accordance with the official method of analysis [28] by drying at 98 °C until a constant weight was reached [29], as well as by the express method using an Evlas-2 apparatus.

### 2.6. Statistical Methods

The experiments were carried out in 3-fold repetition, with analytical measurements in each sample as well. Static processing of the results was carried out using the Microsoft Office Excel 2013 software package, and the arithmetic mean values of the data were found.

Then, using the STATISTICA 13 program, an analysis of variance was carried out with an assessment of the effects and determination of the influence of individual factors based on the regression model.

In the study, analysis of variance was applied, due to the fact that it proves the relationship between the resultant and factor attributes. Regression analysis reveals the relationship of the outcome from the predictors (x1, x2, x3) and also assesses the significance of the model. The estimation of effects demonstrates an assessment of the contributions of factors to the magnitude of the response and their interactions, i.e., they test the significance of the regression coefficients.

## 3. Results

### 3.1. Determination of Lactic Acid Bacteria in Whipped Protein Pastille and Study of Their Probiotic Properties

The study to determine the total number of microorganisms, including LAB, was carried out based on the accredited testing laboratory “Food Safety” of the Almaty Technological University on 14 samples of whipped confectionery products (Table 2). For the reliability of the results, each sample was tested in triplicate.

From all isolated colonies of lactic acid microorganisms, the following strains were identified according to morphological and cultural characteristics. The study of the macromorphology of the studied cultures showed that the cells of LAB had coccal and rod-shaped forms. The diameter of the coccal forms of microorganisms varied from 0.5–0.6 to 1 µm. Some were located singly, in pairs or in the form of chains of various lengths. Rod-shaped bacteria varied in shape, from short, close to coccoid, to long filamentous of various lengths, arranged in chains or singly.

During the study of morphological characteristics, it was revealed that, when determined by the hanging drop method, the cells of the studied cultures were immobile. The results of Gram staining showed that the isolated bacteria can be attributed to lactobacilli, and immobile behaviour is also characteristic of *L. acidophilus* [30]. According to these characteristics, they belong to typical representatives of the LAB family. Table 2 shows data on the identification of isolated strains by morphological features.

The ability of LAB to ferment carbohydrates varies greatly with changing conditions for cultivating microorganisms. In accordance with the *Bergey* determinant, the isolated LAB were assigned to the genus *Lactobacillus*. As a result of the study of cultural and morphological differences, the species affiliation of the strains was determined: all six strains were assigned to *Lactobacillus acidophilus*. Microorganisms of the genus *Lactobacillus* are widespread; some species are the most important representatives of the human microbiota. Previous studies [25] have confirmed that due to the production of organic acids, peroxides and bacteriocins, many strains of lactobacilli exhibit pronounced antagonistic activity against many pathogenic microorganisms. The results of previous works [31,32] confirm that LAB play an important role in the inhibition of food pathogenic and putrefactive microorganisms due to antimicrobial metabolites.

Afterwards, the probiotic properties of the isolated LAB strains were determined. Therefore, their acid-forming ability and resistance to media close to the conditions of the human gastrointestinal tract were determined: acid resistance, bile resistance and salt resistance of LAB (Table 3 and Table 4). The action of probiotics is not limited to the correction of microbiota; according to the results obtained [33], their clinical efficacy is also based on immunomodulatory functions in participation in metabolism.

The pronounced ability to produce lactic acid is one of the best-known biological properties of LAB. These qualities characterize the biochemical and antagonistic activity of LAB. When new bacteriological preparations from lactic acid bacilli are developed, the acid formation activity of microorganisms is also normalized since this parameter also determines their antagonistic activity against other microorganisms. Strains are considered highly active if acid formation in milk or titratable acidity is above 120 °C [31].

The results obtained (Table 3) showed that most of the studied strains had a good capacity for acid formation in the case of M4, M5 and M6 LAB.

To act as a probiotic in the gastrointestinal tract, the bacteria must be able to survive the acidic conditions in the stomach and resist bile acids at the beginning of the small intestine. Approximately 1% of microorganisms in a fermented product survive through the gastrointestinal tract, depending on the strain used and the conditions of administration. To act as a probiotic, the strain must survive in the stomach. The authors of [31] argue that the survival of bacteria in gastric juice depends on their ability to tolerate low pH values. The pH of excreted HCl in the stomach is 0.9; however, after food intake, the pH increases to 3, and after a meal, the stomach empties within 2–4 h [32].

The resistance of the isolated strains to bile is an indirect indicator of cell viability, which may be ensured by their entry into the intestinal zone and the manifestation of probiotic properties. LAB used as probiotics need to survive in the acidic environment of the stomach before entering the colon zone. The adverse effect of gastric juice is the main barrier that bacteria must overcome [33].

All strains isolated grew well in the presence of up to 30% bile concentration. The strains M3 and M4 turned out to be the most resistant to a high concentration of bile (40%) in the substrate (Table 4). LAB isolates were able to tolerate a NaCl concentration of 2%, as shown in Table 4. At a concentration of 4%, the strains M2 and M4 did not grow, and even at a concentration of 6%, strains other than M6 did not grow. NaCl is an inhibitory substance that can inhibit the growth of certain types of bacteria, and consequently if LAB are sensitive to NaCl, then they cannot show their activity in the presence of NaCl. Thus, it was important to check the resistance of isolated LAB to NaCl [34].

Some strains of lactobacilli in stressful situations may show specific metabolic changes that lead to a qualitative or quantitative change in the secreted substances. Strains of the same species can have different intracellular and extracellular substances, as well as a different composition of secreted proteins, as evidenced by previous works [18,35].

*Lactobacillus acidophilus* is often found in the gastrointestinal tract of a healthy person and is resistant to the acidic environment of the stomach and to the action of many antibiotics. Therefore, it provides a longer suppression of pathogenic and opportunistic flora and contribute to the restoration of the intestinal microbiota even during antibiotic therapy. Glucose is fermented with the formation of hydrogen peroxide and lactic and acetic acids. According to the results of the study of the authors [36], this provides a low acidity of the environment, inhibits the growth and reproduction of several pathogenic microorganisms and maintains a healthy balance of the intestinal flora. Meanwhile, a study by the authors in [37] showed that Lactobacillus acidophilus is tolerant to pepsin, an enzyme of gastric juice.

According to other authors [38], proteins associated with the surface layer of *L. acidophilus* have an immunomodulatory effect, increase the release of antibodies and cause phagocytosis, which reduces the number of pathogens of upper respiratory tract infection. Based on the studies carried out, it can be concluded that the LAB found in whipped protein pastille belong to the genus *L. acidophilus*.

### 3.2. Influence of Technological Factors on the Growth and Development of Lactic Acid Microorganisms by the Method of Mathematical Modelling

To substantiate the influence of technological factors on the growth and development of lactic acid microorganisms, samples from No. 1 to No. 9 were studied for mathematical modelling. Table 5 shows a full-factor plan of a 2**(3-0) experiment with three independent variables (drying duration, drying temperature, amount of introduced probiotic starter culture) and the results of experiments to determine the level of penetration, humidity and the number of LAB.

According to Table 5, it can be concluded that the level of penetration directly depends on the amount of ferment introduced. Therefore, an increase in the amount of starter increases the level of penetration, which in turn leads to changes in the rheological properties of the product—the friability of the product increases. Therefore, it is necessary to maintain the optimal amount of introduced starter, since with a further increase in the dose of introduced starter and the level of penetration, the necessary structure of the product may be lost. Additionally, the humidity and temperature of product processing is of great importance in preserving the properties of probiotic starter cultures. At a high humidity index Y_2_, the largest number of lactic acid microorganisms is preserved, and when the temperature X_2_ decreases from 70 °C to 50 °C, the probability of preserving the largest number of lactic acid microorganisms Y_3_ increases. According to this study, it can be concluded that the optimal technological solution for preserving a larger number of lactic acid microorganisms is to reduce the processing temperature, increase the time and introduce the optimal amount of dry probiotic sourdough.

Figure 1 and Figure 2 show fitted surfaces for Y_1_ (penetration level) demonstrating the influence of technological factors such as the drying duration and temperature and the dosage of the probiotic starter.

As can be seen from Figure 1, the higher the temperature and longer the drying process, the lower the penetration level becomes. The volume and texture of the products become denser. Figure 2 shows that the higher the dosage of probiotic inoculum and the shorter the drying time, the higher the penetration level becomes, and the product becomes crumblier. Therefore, more than 0.021% of probiotic starter should not be added to protein pastille, as the rheological and organoleptic properties of finished products will deteriorate.

After constructing the experimental plan and conducting the necessary studies, an analysis of the evaluation of the effects was performed. Analysis of the evaluation of effects allowed us to identify the influence of independent changes (factors) on the level of penetration, which is one of the main indicators of the rheological properties of whipped protein pastille. The results of the analysis regarding the evaluation of the effects are given in Table 6.

Table 6 shows that the level of penetration is influenced by all three factors and the interaction between the first and third factors (*p* < 0.05).

Additionally, based on the analysis of the effects assessment, coefficients were obtained for constructing a linear regression equation:Y_1_ = 21.44 − 2.25x_1_ − 1.84x_2_ + 3.42x_3_ − 2.11x_1_x_3_

The adequacy of this linear equation is evidenced by the coefficient of determination R² = 0.83. It was substituted using the initial values for x_1_, x_2_ and x_3_ to obtain the following values of Y_1_:

У_1_ = 21.44 − 2.25 × 12 − 1.84 × 50 + 3.42 × 0 − 0.35 × 50 × 0 = −97.56 for the minimum value

У_1_ = 21.44 − 2.25 × 16 − 1.84 × 70 + 3.42 × 0.07 − 0.35 × 70 × 0.07 = −144.84 for the maximum value

The sign “−” at the coefficient indicates that with the increase in the factor value the response value (penetration level) decreases. The equation shows that the factor x₃ (amount of inoculum), X_1_ (duration of drying) and the combination of factors X_1_X_3_ (duration and amount of inoculum) have the strongest influence, as they have the highest absolute value of coefficients. The combination of factors X_1_X_2_ (duration and drying temperature) and X_2_X_3_ (drying temperature and amount of inoculum applied) have the least influence according to Table 6.

To determine the influence of technological factors, such as the duration and temperature of drying and the amount of probiotic starter introduced, on the moisture content of finished products, a regression and variance analysis was carried out (Table 7 and Table 8), within which coefficients were obtained for constructing a regression equation with a determination coefficient R^2^ = 0.88.

Regression analysis shows the influence of the predictors X_1_, X_2_, X_3_ on the resultant attribute Y_2_ and the identification of the regression model (equations).

Regression equation: y_2_ = 110.41 − 4.74x_1_ − 1.13x_2_

y_2_ = 110.41 − 4.74 × 12 − 1.13 × 50 = −2.97 for the minimum value

y_2_ = 110.41 − 4.74 × 16 − 1.13 × 70 = −44.53 for the maximum value

The regression equation clearly demonstrates that the more processing takes place, the lower the moisture content of the finished products will be, despite the presence of any amount of probiotics in its composition.

From Table 8, it can be concluded that the moisture content of the finished products is influenced by two factors—temperature and drying time, which is clearly demonstrated in Figure 3. The moisture content of the products is higher when the processing temperature is 50 °C and the processing time is 12 h. To avoid the formation of a dry crust on the surface of the product, which is formed because of the uneven distribution of moisture, and to achieve the necessary rheological properties of the finished product, the optimal technological parameters were selected, considering the influence of all three factors.

### 3.3. Influence of Technological Factors on the Content of Lactic Acid Organisms in Whipped Protein Pastille

To determine the effect of the duration and temperature of drying and the amount of introduced probiotic starter culture on the content of lactic acid microorganisms in finished products, an analysis was carried out to evaluate the effects (Table 9). As part of the analysis of the effects assessment, coefficients were obtained for constructing a regression equation with a determination coefficient R² = 0.96, confirming the good dependence of the variables:Y_3_ = 4 − 0.84x_2_ + 4.21x_3_

Y_3_ = 4 + 4.21 × 0 = 4 for the minimum value

Y_3_ = 4 + 4.21 × 0.07 = 4.29 for the maximum value

Table 9 shows that the factor affecting the mould is only the dosage of probiotic starter applied. However, this study was conducted with a control sample with no inoculum. If we analyse without a control sample, only between the minimum and maximum dosage, then there will be another significant factor affecting the survival of probiotics and this factor is temperature.

From Table 9, it can be concluded that the LAB content in whipped protein pastille is strongly influenced by factors such as the amount of dry probiotic starter added. At the same time, an X_2_ factor coefficient equal to −0.04 indicates a negative impact on the content of LAB; that is, the higher the processing temperature is, the lower the content of LAB. Moreover, the X_3_ coefficient equal to 4.21 indicates a positive effect; that is, the more starter is introduced, the higher the content of LAB will be (Figure 4). However, it is not worth adding more than 0.7 mg of starter, as the rheological properties of the finished products will deteriorate.

Based on the analysis of the effects assessment and Figure 2, the best indicator for the content of LAB is demonstrated by Sample No. 6 (10 CFU), and the smallest amount is shown by Sample No. 9 (2–3 CFU).

Figure 4 shows that the best indicator of the content of LAB (10 CFU) was the sample with the introduction of probiotic in the amount of 0.021% of culture, with a drying time of 16 h at 50 °C. According to this study, it can be concluded that the optimal technological solution to preserve more LAB is to reduce the processing temperature, increase the time and apply the optimal amount of dry probiotic starter. At the same time, depending on the increase in the amount of probiotic culture added to the composition, the penetration number decreases and the moisture content increases.

Thus, the study revealed the optimal technological mode to produce this type of pastille product, namely the drying of the pastille mass in a dehydrator at a temperature of 50 °C for 16 h. This mode of heat treatment preserves a greater number of lactic acid bacteria, with stable structural-mechanical, physico-chemical and organoleptic properties of the protein pastille. The resulting functional product can be characterized as synbiotic, as the main raw material of plant origin contains a large amount of fibre, which acts as a prebiotic, and the strain of the microorganism, which acts as a probiotic. In synergy, this gives a unique product, with the properties of both components aimed at maintaining immune status.

The most common and thoroughly studied probiotic cultures are LAB. The authors of [2] provide information that the genus *Lactobacillus* is not only one of the most available on the market but also safe for use, in most cases resistant to antimicrobial agents. This is confirmed by the Qualified Presumption of Safety (QPS) status, as recommended by the European Food Safety Authority (EFSA). Of scientific interest is the species *L. acidophilus*, a Gram-positive LAB that has recently been widely used as a probiotic supplement.

A previous work [39] provided data on the probiotic *L. acidophilus*, which plays an important role in many aspects of human health, including the regulation of the balance of intestinal flora, lowering cholesterol levels, modulating immunity and counteracting cancer. Compared to many other probiotics, *L. acidophilus* has a better resistance to both acid and bile salts. These characteristics facilitate the survival and reproduction of *L. acidophilus* in gastrointestinal conditions.

It was also proposed [40] that *L. acidophilus* is an effective and safe method in the treatment of diarrhoea and even in the presence of immunosuppression. The advantage of using lyophilized organisms as probiotics is described. In addition, the use of even heat-treated *L. acidophilus* has been shown to induce protection against *Candida albicans* in immunocompromised mice. In accordance with this study, the problem of the survival of *L. acidophilus* in foods with high heat treatment can be solved.

According to [41], a component that is planned to be added to a product due to functionality studies can completely change the texture of the product or its sensory characteristics. In such cases, as in other food products, the colour, flavour and texture of soft confectionery products are critical factors for consumer decision making and the success of these products [42]. Therefore, in parallel to the determination of the identity of LAB the effect of three factors (drying duration, drying temperature, amount of probiotic starter introduced) on the penetration level, moisture content and number of lactic acid microorganisms after processing was studied. The penetration level can be used to judge the rheological properties of pastille.

A disadvantage of the study is that experiments were conducted on a certain type of raw material from one batch. There is a possibility that if a different batch of raw materials is used, the number of lactic acid microorganisms may vary. Consequently, it will be necessary to study the preservation of lactic acid microorganisms, as well as to study the effect of exposure to different types and varieties of fruit and berry raw materials on the rheological and organoleptic parameters of the finished products.

The advantage of the study is that we proved the survival of probiotic cultures in the developed whipped protein pastille, which can provide a carrier of probiotics as an alternative to dairy products. The optimal dosage of the probiotic was determined, which allowed us to enrich the product and at the same time to maintain the proper rheological and organoleptic properties necessary for industrial production.

The limitation of the study is that there remains a need to further investigate the persistence of LAB in the developed product, i.e., to study more thoroughly how the lactic acid bacteria manifest themselves after 3, 6 and 9 months of storage. In this regard, further studies are being carried out to determine the amount of LAB depending on the storage period.

Our research will provide a basis for further study of the relationship between probiotics and medicinal herbs in confectionery products in synergy and their influence on the release of biologically active substances in the product during digestion through in vitro studies. These studies will make it possible to expand the range of functional foods, with a vitaminising and tonic effect to strengthen the immune system of the body.

## 4. Conclusions

In this study, it was found that bile protein marshmallow can serve as a carrier for probiotic enrichment and be an alternative to dairy products, as it was found that microorganisms of the genus *L. acidophilus* survived after processing treatment and *L. acidophilus* M3 and *L. acidophilus* M4 strains were the most resistant to a high bile concentration in the substrate. The optimal technological mode of preparation of Belevskaya whipped protein pastille (drying the pastille mass in a dehydrator at a temperature of 50 °C for 16 h) and the optimal dosage of probiotic (0.021%), which allowed us to enrich the product with LAB and preserve the organoleptic and rheological properties of finished products, were revealed. Based on the analysis of variance and the regression equations obtained, it was found that the growth of LAB in the product was strongly influenced by the amount of sourdough starter applied (coefficient of determination R² = 0.96). The penetration level was influenced by factors such as the amount of introduced probiotic starter, drying time and the interaction between the factors of drying time and the amount of introduced starter.

In general, the results of the study showed the effect of probiotic cultures on the rheological properties of protein pastilles; more than the established dosage should not be applied, as the higher the dosage, the product becomes crumblier and more friable, which will optimise the processes of industrial production. This study provides an opportunity to use the obtained data on a larger scale and is applicable for other products similar in technological process.

## Figures and Tables

**Figure 1 foods-12-03700-f001:**
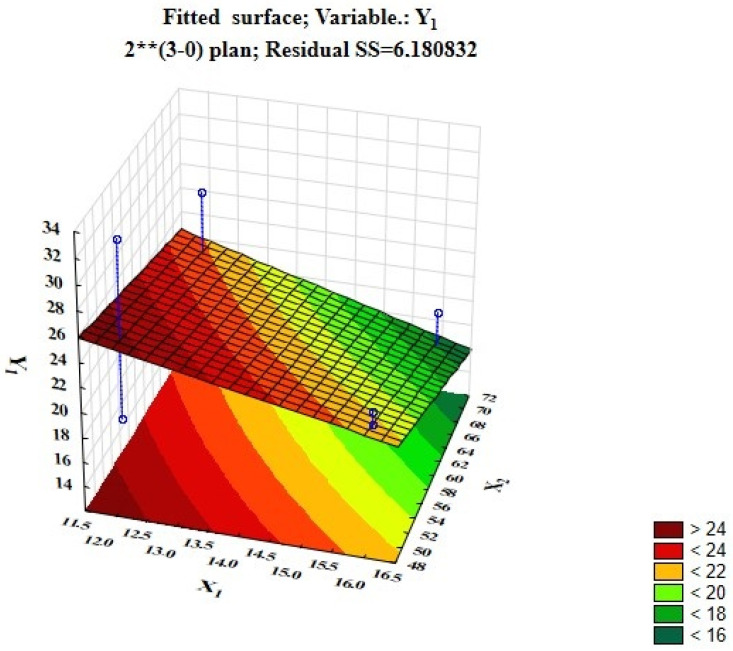
Fitted response surface demonstrating the dependence of penetration on the drying time and treatment temperature.

**Figure 2 foods-12-03700-f002:**
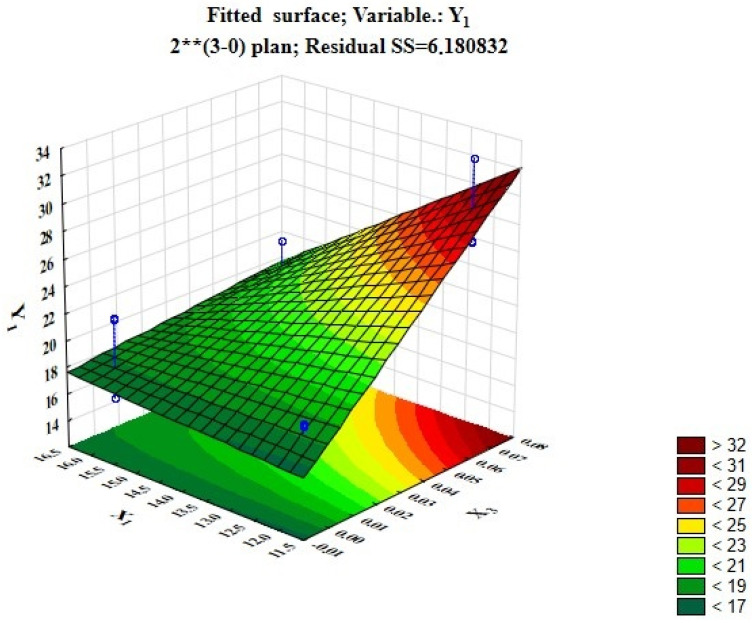
Fitted response surface demonstrating the dependence of penetration on the drying time and dosage of probiotic starter applied.

**Figure 3 foods-12-03700-f003:**
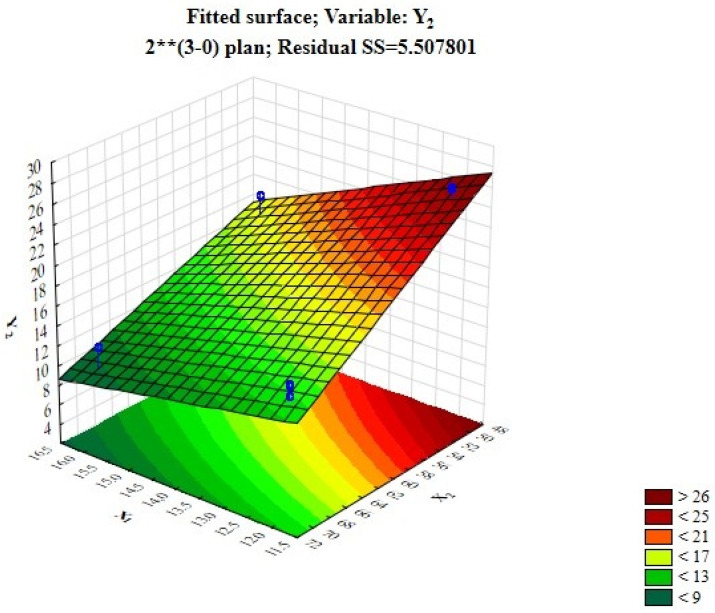
Fitted response surface demonstrating the effect of the dry probiotic starter amount and drying temperature on whipped protein pastille moisture.

**Figure 4 foods-12-03700-f004:**
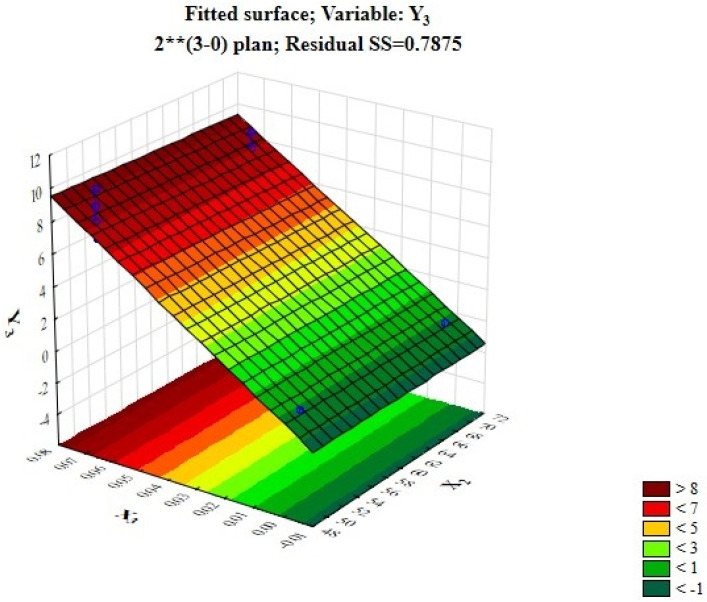
Fitted response surface demonstrating the effect of the dry probiotic sourdough amount and drying time on the content of lactic acid organisms in whipped protein pastille.

**Table 1 foods-12-03700-t001:** Prototypes of Belevskaya whipped protein pastilles developed.

Sample No.	(X_3_) Probiotic Dosage (g)	(X_1_) Drying Time (h)	(X_2_) Drying Temperature (°C)
1	NP	12	50
2	NP	16	50
3	NP	12	70
4	NP	16	70
5	0.07 g (0.021%)	12	50
6	0.07 g (0.021%)	16	50
7	0.07 g (0.021%)	12	70
8	0.07 g (0.021%)	16	70
9	0.035 g (0.011%)	14	60
10	0.03 g (0.009%)	12	50
11	0.03 g (0.009%)	16	50
12	0.03 g (0.009%)	12	70
13	0.03 g (0.009%)	16	70
14	50.05 g (0.015%)	14	60

NP: No probiotic.

**Table 2 foods-12-03700-t002:** Morphological features of lactic acid bacteria isolate from whipped protein pastilles.

Designation of Strains	Morphological Features of Microorganism Cells	Bacteria Strains
M1	Single and in the form of chains of sticks 12–17 µm long and 1.2–1.6 µm thick.	*L. acidophilus* M1
M2	In the form of chains of 3–5 rod-shaped segments, 10–15 µm long, 0.8–1.0 µm.	*L. acidophilus* M2
M3	In the form of chains of 3–9 rod segments 15–25 µm long and 1.2–1.5 µm thick.	*L. acidophilus* M3
M4	Chains of 5–9 segments, cell size, length 10–15 µm, width 1.3–14 µm.	*L. acidophilus* M4
M5	Bacteria in a chain of 3–4 segments, cell size 10–15 µm, 1.0–1.3 µm.	*L. acidophilus* M5
M6	Short chains of 4–6 segments, length 10–12 µm, width 1.2–1.4 µm.	*L. acidophilus* M6

**Table 3 foods-12-03700-t003:** Determination of acid formation in milk in isolated strains.

Strain	Acid Formation in Milk, °T	
	After 8 h	Limit
M1	85	115
M2	80	110
M3	100	120
M4	80	180
M5	145	180
M6	90	240

**Table 4 foods-12-03700-t004:** Determination of bile and NaCl resistance of lactic acid bacteria.

Designation of Strains	pH Values			Growth in Hydrolysed Milk, with Bile Content, %			Growth in Hydrolysed Milk, WITH NaCl Content, %		
	6.5	5.5	4.5	20	30	40	2	4	6
M1	+	+	−	+	+	−	+	+	−
M2	+	+	−	+	+	−	+	−	−
M3	+	+	+	+	+	+	+	+	−
M4	+	+	+	+	+	+	+	−	−
M5	+	+	+	+	+	−	+	+	−
M6	+	+	+	+	+	−	+	+	+

“+”: remain in the active phase; “−”: out of stock.

**Table 5 foods-12-03700-t005:** Plan and results of the experiment.

Plan Preparation	Plan 2**(3-0) Plan (3**(3-1) fractional Factorial Design							
	*n*	Center	X_1_	X_2_	X_3_	Y_1_	Y_2_	Y_3_
9 (C)	1	0	14	60	0.035	16.24	11	3
12	2	1	12	70	0	18.8	15.1	0
3	1	1	12	70	0	18.7	15	0
15	2	1	16	50	0.07	22.49	19.3	8
22	3	1	16	70	0	15.55	9.1	0
1	1	1	12	50	0	18.85	26.1	0
2	1	1	16	50	0	21.51	17	0
25	3	1	12	70	0.07	26.86	13.9	8
13	2	1	16	70	0	15.57	9.2	0
20	3	1	16	50	0	21.49	16.8	0
21	3	1	12	70	0	18.9	15.2	0
27 (C)	3	0	14	60	0.035	16.22	10.8	2
8	1	1	16	70	0.07	19.79	11.9	8
16	2	1	12	70	0.07	26.85	14	9
18 (C)	2	0	14	60	0.035	16.23	10.9	2
24	3	1	16	50	0.07	22.5	19.5	9
5	1	1	12	50	0.07	32.89	25.9	7
19	3	1	12	50	0	18.83	25.9	0
26	3	1	16	70	0.07	19.81	12	8
23	3	1	12	50	0.07	32.88	25.8	9
10	2	1	12	50	0	18.84	26	0
6	1	1	16	50	0.07	22.51	19.4	10
17	2	1	16	70	0.07	19.78	12	8
11	2	1	16	50	0	21.5	16.9	0
14	2	1	12	50	0.07	32.9	26	8
7	1	1	12	70	0.07	26.84	14.1	9
4	1	1	16	70	0	15.56	9.2	0

X_1_: drying time (h); X_2_: temperature (°C); X_3_: amount of starter (mg); Y_1_: penetration (mm); Y_2_: moisture content (%); Y_3_: lactic acid microorganisms (colony forming units).

**Table 6 foods-12-03700-t006:** Evaluation of effects on the level of penetration of whipped protein pastille.

Factor	Effect Estimates; R–q. = 0.83; Velocity 0.77649 (Design: 2**(3-0) Design (3**(3-1) Fractional Factorial Design 2**(3-0) Design; Residual (SS = Sum of Squared Deviations) = 6.180832 SV(Scheduled Variable) Y1									
	Effect	SE	t (20)	*p*	Coefficient	>95% CL	Coefficient	SE	<95% CL	>95% CL
Medium/Free Member	21.44	0.478	44.81	<0.00001	20.44	22.44	21.44	0.48	20.44	22.44
(1)X_1_	−4.51	1.015	−4.44	0.00025	−6.62	−2.39	−2.25	0.51	−3.31	−1.19
(2)X_2_	−3.68	1.015	−3.63	0.00167	−5.79	−1.56	−1.84	0.51	−2.89	−0.78
(3)X_3_	6.83	1.015	6.73	0.000002	4.72	8.95	3.42	0.51	2.36	4.47
1 on 2	−0.64	1.015	−0.63	0.53441	−2.76	1.47	−0.32	0.51	−1.38	0.74
1 on 3	−4.22	1.015	−4.15	0.00049	−6.33	−2.09	−2.11	0.51	−3.17	−1.05
2 on 3	−0.69	1.015	−0.68	0.50338	−2.81	1.42	−0.35	0.51	−1.40	0.71

CL: Confidence limit; ST: standard error coefficient.

**Table 7 foods-12-03700-t007:** Regression analysis to determine the influence of technological factors, such as the duration and temperature of drying and the amount of probiotic starter, on the moisture content of finished products.

Factor	Regression; R-sq. = 0.88; Velocity 0.84 Plan: 2**(3-0) Residual (SS—Sum of Squared Deviations) = 5.5078 SV(Scheduled Variable)Y2					
	Regression Coeficient	ST	t(20)	*p*	<95% CL	>95% CL
Medium/Free Member	110.41	21.07	5.24	0.00004	66.45	154.38
(1)X1	−4.74	1.47	−3.21	0.00438	−7.82	−1.66
(2)X2	−1.13	0.34	−3.31	0.00346	−1.85	−0.42
(3)X3	−132.85	126.93	−1.05	0.30772	−397.63	131.91
1 нa 2	0.05	0.02	1.99	0.06022	−0.002	0.09
1 нa 3	11.61	6.84	1.69	0.10538	−2.67	25.88
2 нa 3	−0.25	1.37	−0.18	0.85691	−3.11	2.61

CL: Confidence limit; ST: standard error coefficient.

**Table 8 foods-12-03700-t008:** Dispersion analysis of variance for full factorial design 3**(3-1).

Factor	Dispersion Analysis; Variable: Y2; R-Square = 0.88; Speed 0.84 (Plan: 2**(3-0) plan (3**(3-1) Fractional Factorial Plan) 2**(3-0) Plan; Residual (SS—Sum of Squared Deviations) = 5.5078 SV(Scheduled Variable) Y2				
	SS	Cc	MS	F	*p*
(1)X_1_	208.270	1	208.270	37.813	0.000005
(2)X_2_	540.550	1	540.550	98.143	<0.000001
(3)X_3_	6.303	1	6.304	1.144	0.297443
1 on 2	21.850	1	21.850	3.967	0.060228
1 on 3	15.844	1	15.844	2.877	0.105387
2 on 3	0.184	1	0.184	0.033	0.856910
Error	110.156	20	5.508	–	–
General SS	903.159	26	–	–	–

**Table 9 foods-12-03700-t009:** Evaluation of effects on the content of lactic acid organisms in whipped protein pastille.

Factor	Effect Estimates; R–q. = 0.96; Velocity 0.95 (Design: 2**(3-0) design (3**(3-0) Fractional Factorial Design 2**(3-0) Design; Residual (SS = Sum of Squared Deviations) = 0.7875 RFP Y3									
	Effect	SE	t (20)	*p*	Coefficient	>95% CL	Coefficient	SE	<95% CL	>95% CL
Medium/Free member	4.00	0.17	23.42	<0.00001	3.64	4.35	4.00	0.17	3.64	4.35
(1)X_1_	0.08	0.36	0.23	0.8234	−0.67	0.84	0.04	0.18	−0.34	0.42
(2)X_2_	−0.08	0.36	−0.23	0.8265	−0.84	0.67	−0.04	0.18	−0.42	0.34
(3)X_3_	8.42	0.36	23.23	<0.00001	7.66	9.17	4.21	0.18	3.83	4.58
X_1_ X_2_	−0.42	0.36	−1.15	0.2645	−1.17	0.34	−0.21	0.18	−0.58	0.17
X_1_ X_3_	0.08	0.36	0.23	0.8212	−0.67	0.84	0.04	0.18	−0.34	0.47
X_2_ X_3_	−0.08	0.36	−0.23	0.8232	−0.84	0.67	−0.04	0.18	−0.42	0.34

CL: Confidence limit; ST: standard error coefficient.

## Data Availability

The data presented in this study are available on request from the corresponding author. The data are not publicly available due to confidentiality agreements.

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
