# Peer review of "Development of a Technology for Protein-Based, Glueless Belevskaya Pastille with Study of the Impact of Probiotic Sourdough Dosage and Technological Parameters on Its Rheological Properties"

_foods, 2023, doi:10.3390/foods12193700_

Round 1

Reviewer 1 Report

The purpose of the article and its significance are not stated clearly.

The abstract should include a final paragraph describing the most relevant conclusions. Please clearly state the implications for research, practice, and society in the conclusions or in the abstract. I suggest mentioning the originality of the study and the novelty it brings to the field.

The introduction section could be more comprehensive.

Please, briefly add future perspectives and further applied applications of this specific research work in the discussion section before the conclusion.

I suggest including information regarding the advantages and limitations of the study.

Please state the conclusions clearly and try to avoid repetitions from the discussion section.

Author Response

Thank you for your time and your recommendations for improving our article. Your comments are very valuable to us. After all, they will help us to make our article better.

We have prepared answers to your questions:

  1. The purpose of the article and its significance are not stated clearly.

We have edited the aim of our study, we hope that it has become clearer.

In order to identify the dependence of technological factors on the rheological properties of the product and the growth of lactic acid microorganisms in battered confectionery products we set 3 objectives:

  1. It is necessary to determine the optimal dosage of probiotic starter, which would allow to enrich the product with probiotics and, at the same time, would not worsen the rheological and organoleptic properties of finished products;
  2. The number of surviving lactic acid bacteria in bunched pastilles after processing treatment should be investigated to determine the effectiveness of their probiotic properties and stability in acidic medium;
  3. The influence of technological factors on the rheological properties of pastilla, growth and development of lactic acid microorganisms should be substantiated using mathematical modelling, revealing the significance of factors by means of dispersion and regression analysis.
  1. The abstract should include a final paragraph describing the most relevant conclusions. Please clearly state the implications for research, practice, and society in the conclusions or in the abstract. I suggest mentioning the originality of the study and the novelty it brings to the field.

We have edited the abstract and conclusions of our studies. In the abstract section, it was included the paragraph:

“The proper functioning of the gastrointestinal tract plays an important role in strengthening the immune system. It is an undeniable fact that lactic acid microorganisms are necessary for the proper functioning of the gastrointestinal tract, the source of which are mainly dairy products. However, there is a problem with the digestibility of lactose; therefore, alternative sources and carriers of probiotics are of particular interest. Due to its dietary and natural properties, protein marshmallow can serve as such a carrier. Therefore, the direction of this study is to identify the dependence of technological factors on the rheological properties of the product and the growth of lactic acid microorganisms in confectionery products enriched with lyophilized strains. According to the results of the study, the following were determined.”

Additionally, the conclusions section was re-organized and the final version is as shown:

“In this study, it was found that bile protein marshmallow can serve as a carrier for probiotic enrichment and be an alternative to dairy products, as it was found that micro-organisms of the genus L. acidophilus survived after processing treatment and that L. aci-dophilus M3 and L. acidophilus M4 strains were the most resistant to high bile concentra-tions in the substrate. The optimal technological mode of preparation of protein pastille (drying the pastille mass in a dehydrator at a temperature of 50 °C for 16 hours.) and the optimal dosage of probiotic (0.021%), which allowed enrichment of the product with lactic acid bacteria and preserved the organoleptic and rheological properties of the finished products, were revealed. Based on the analysis of variance and the regression equations obtained, it was found that the growth of LAB in the product was strongly influenced by the amount of sourdough starter applied (coefficient of determination R² = 0.96). The pene-tration level was influenced by factors such as the amount of introduced probiotic starter, drying time and the interaction between the factors of drying time and the amount of in-troduced starter.

In general, the results of the study showed the effect of probiotic cultures on the rheo-logical properties of protein pastilles, more than the established dosage should not be ap-plied, as the higher the dosage, the more crumbly and friable the product becomes, which will optimize the processes of industrial production. This study provides an opportunity to use the obtained data on a larger scale and is applicable for other products similar in technological process.”

  1. The introduction section could be more comprehensive.

We've expanded the introductory section. In concrete, it was added the following paragraphs:

“It is also noted that in the original version of the recipe, there were no sweeteners; only lat-er, to increase the sweetness was honey added to the pastille mass, which was then re-placed with sugar [3].

The main raw material for the production of whipped protein pastilles is apple puree, which is obtained from baked apples, in which a special form of fiber is pectin. The action of pectin is more pronounced during the heat treatment of apples, and this process is ac-companied by the acquisition of a gel-like sheen on the surface of apples. It is known that the use of applesauce as the main prescription component carries a number of positive aspects. Emerging evidence suggests that pectin may help repair and preserve the intesti-nal mucosa [4]. Pectin also helps in the modulation of intestinal bacteria, eliminates tox-ins in the intestines and reduces inflammation [5].

A study was conducted showing that the chemical and nutritional properties of apples from 12 varieties of apples manufactured industrially (heat treatment at 95 °C for 2 minutes, pasteurization at 90 °C for 5 minutes, and cooling at 20 °C for 20 minutes) are very close to those of fresh apple pulp [6].

The topic of probiotic products is an integral part of the nutrition industry. Eating confec-tionery products produced using lactic acid bacteria (LAB) contribute to positive changes in the intestinal microbiota.

The results of the authors' research [7] showed that, despite some loss of probiotic adhe-sion, the combined presence of extract and probiotic is more effective in reducing the over-all amount of adhered viable pathogen cells than the probiotic alone, regardless of the probiotic/pathogen system considered.

A review article [8] shows that the potential of probiotics is to increase innate and ac-quired immunity and activate anti-inflammatory and anti-allergic effects. This study shows the relevance of the use of probiotics in human life. However, the question remains unresolved regarding how easy and affordable it is to consume them in everyday life, par-ticularly for people suffering from the indigestibility of milk protein.

In [9], the authors proved the possibility of producing probiotic grape marmalade without the use of dairy products.

Previous works [10] studied the influence of technological parameters in the production of marmalade products on the safety of vital nutrients. In their further works [24], the mor-phology of the probiotic culture was studied, and it was found that the volume of whey (250 ml) affects the growth of LAB in the marmalade product. The optimal amount of pro-biotic culture to be added to the serum was 0.01–0.02 g, while the time of reviving the mi-croorganisms was 6 hours. As a result of the study, the confectionery product contained from 1 to 3 CFU/g lactic acid culture and 1.7 and 2.2 times more antioxidants compared to the control, which justifies the benefits for the gastrointestinal tract and maintaining hu-man immunity due to the production of its own interferon.

The studies conducted in this area prove the promise of this direction; however, it is nec-essary before previous to industrial applicability, it is necessary to study technological methods for introducing a probiotic culture as an active enriching ingredient, followed by a study of the safety of LAB after technological processing and storage. It is also necessary to analyze exactly what effect the dosage and type of probiotic culture has on the rheologi-cal properties of finished products. [11].

To identify the dependence of technological factors on the rheological properties of the product and the growth of LAB in battered confectionery products, it was set 3 objectives:

  1. It is necessary to determine the optimal dosage of probiotic starter, which would allow enrichment of the product with probiotics and, at the same time, would not worsen the rheological and organoleptic properties of finished products;
  2. The number of surviving LAB in Belevskaya whipped protein pastilles after processing treatment should be investigated to determine the effectiveness of their probiotic properties and stability in acidic medium;
  3. The influence of technological factors on the rheological properties of Belevskaya whipped protein pastilles and the growth and development of LAB should be substantiat-ed using mathematical modeling, revealing the significance of factors by means of disper-sion and regression analysis.
  1. Please, briefly add future perspectives and further applied applications of this specific research work in the discussion section before the conclusion.

In the discussion section, we have added the perspectives and further applications of this particular research paper. In the revised version of the manuscript, it was added the following paragraph:

“The advantage of the study is that we proved the survival of probiotic cultures in the developed whipped protein pastille, which can provide a carrier of probiotics as an alternative to dairy products. The optimal dosage of the probiotic was determined, which al-lowed enrichment of the product and maintenance of the proper rheological and organoleptic properties necessary for industrial production.”

  1. I suggest including information regarding the advantages and limitations of the study.

We modified this section, adding the benefits of our experiment (lines) and the limitations of the study (lines). In the corrected version of the manuscript, it is specified that:

“The limitation of the study is that there remains a need to further investigate the persistence of LAB in the developed product. To study more thoroughly how lactic acid bacteria manifest themselves after 3, 6 and 9 months of storage. Thus, further studies are being carried out to determine the amount of LAB depending on the storage period.”

  1. Please state the conclusions clearly and try to avoid repetitions from the discussion section.

We have changed the conclusions, hopefully they are clearer. In concrete, the conclusions were stated as follows:

“In this study, it was found that bile protein marshmallow can serve as a carrier for probiotic enrichment and be an alternative to dairy products, as it was found that micro-organisms of the genus L. acidophilus survived after processing treatment and that L. acidophilus M3 and L. acidophilus M4 strains were the most resistant to high bile concentrations in the substrate. The optimal technological mode of preparation of protein pastille (drying the pastille mass in a dehydrator at a temperature of 50 °C for 16 hours.) and the optimal dosage of probiotic (0.021%), which allowed enrichment of the product with lactic acid bacteria and preserved the organoleptic and rheological properties of the finished products, were revealed. Based on the analysis of variance and the regression equations obtained, it was found that the growth of LAB in the product was strongly influenced by the amount of sourdough starter applied (coefficient of determination R² = 0.96). The penetration level was influenced by factors such as the amount of introduced probiotic starter, drying time and the interaction between the factors of drying time and the amount of introduced starter.

In general, the results of the study showed the effect of probiotic cultures on the rheo-logical properties of protein pastilles, more than the established dosage should not be ap-plied, as the higher the dosage, the more crumbly and friable the product becomes, which will optimize the processes of industrial production. This study provides an opportunity to use the obtained data on a larger scale and is applicable for other products similar in technological process.”

Thank you again for your comments, we hope we have answered all your questions. Best regards, authors.

Reviewer 2 Report

After reading this manuscript, I have some questions to ask.

 1. Line 107: Please add city name in between company and country names.

 2. Line 110: 102-106? 2 and 6 should be superscripted, right?

 3. Line 114-117: In this paragraph, I am confused because the authors put test tube at 6 degree Cesius, for 30, 60, and 90 mins. So the first sentence ‘heat resistance’ may be considered as ‘cold resistance’. Also, in Table 3, after 8 hours treatmet at 85, 80, 100, 80, 145, and 90 degree Cesius, I wonder where is the material and methods section for this experiment?

 4. In addition, the authors measured acid formation in Table 3. What kind of acid is measured, also I cannot locate the procedure in material and method section

 5. Line 124, I understand the optimum growth pH of Lactobacillus acidophilus is around 6.5. We also knew that Lactobacillus acidophilus can survive at low pH condition even pH 4.0 for couple hours? Also in Table 4, M3-M4 can survive in pH 4.5 treatment. Do authors consider to test it at even lower pH?

 6. Lines 144 and 145: Please add city and state names in between company and country names.

 7. Page 6, lines 202-203, lines 212-214, and line 227-228: One sentence as one paragraph. A mind suggestion, the authors can add more description or combine those sentences with other paragraphes.

 8. Fig 1 and Fig 2: Please increase resolutions of both pictures. Can the authors use real parameters to replace X1, X2, and Y1/Y3.

Author Response

Thank you for your time and your recommendations for improving our article. Your comments are very valuable to us. After all, they will help us to make our article better.

We have prepared answers to your questions:

  1. Line 107: Please add city name in between company and country names.

We have examined the packaging of the agar used and found that it indicates Bhiwadi, corrected and added Bhiwadi, India and we have added this clarification to the article in line 160 (lines shifted as we were making changes).

  1. Line 110: 102-106? 2 and 6 should be superscripted, right?

Yes you are correct, those are superscript numbers (degrees), we have corrected that error. (line 164 of the original version of the manuscript)

  1. Line 114-117: In this paragraph, I am confused because the authors put test tube at 6 degree Cesius, for 30, 60, and 90 mins. So the first sentence ‘heat resistance’ may be considered as ‘cold resistance’. Also, in Table 3, after 8 hours treatmet at 85, 80, 100, 80, 145, and 90 degree Cesius, I wonder where is the material and methods section for this experiment?

We made a mistake (typo), in this case it should have been 60 ºC, we have corrected this error in the article (line 170 of the original version of the manuscript)

  1. In addition, the authors measured acid formation in Table 3. What kind of acid is measured, also I cannot locate the procedure in material and method section

We supplemented this section in the article and described the methodology. The acid-forming capacity of the cultures under study was determined by the rate of lactic acid formation per unit time.  There was a translation error in Table 3, we corrected it to Turner's Degree.

In the materials and methods section, it was added the following paragraph:

“The acid-forming activity of LAB was determined by examining the acid-forming capacity and limiting acid formation. The acid-forming capacity of the cultures under study was determined by the rate of lactic acid formation per unit time (1 h). To determine the acid formation activity, isolates were seeded in 0.3 ml tubes with 10 ml of milk. The tubes were placed in a thermostat at 37 °C. After 24 hours, acidity was determined by the titrimetric method. For this purpose, 10 ml of the sample was diluted with 20 ml of distilled water, and 1-2 drops of phenolphthalein indicator were added. Titration was carried out with 0.1 n NaOH solution until the appearance of a stable pink color. Limit acid formation was determined by sowing 0.3 ml of the tested culture in 10 ml of skim milk. The cultures were thermostated at 37 °C. After 7 days, titratable acidity expressed in Turner degrees was determined [24].”

  1. Line 124, I understand the optimum growth pH of Lactobacillus acidophilus is around 6.5. We also knew that Lactobacillus acidophilus can survive at low pH condition even pH 4.0 for couple hours? Also in Table 4, M3-M4 can survive in pH 4.5 treatment. Do authors consider to test it at even lower pH?

Unfortunately, we did not carry out a study below pH 4.5, as the lactic acid bacteria survive longer in the product during storage in a stable state at pH 4.5. We will take this into consideration and conduct an additional study, the results of which we will publish as an addendum in the next article.

Experiments were carried out to determine resistance to hydrochloric acid. Hydrochloric acid was added to the liquid MRS medium inoculated with the tested bacteria. The pH values of the medium were set from 4.5 to 6.5. As a result, it was found that the bulk of the isolated strains had convincing resistance to acid medium. Only strains M1, M2 showed low growth when hydrochloric acid pH 4.5 was added.

  1. Lines 144 and 145: Please add city and state names in between company and country names.

We have added a city and state, for the stats programme it is California for the xcel programme it is Lemogyne, Penn State (lines 209-210)

  1. Page 6, lines 202-203, lines 212-214, and line 227-228: One sentence as one paragraph. A mind suggestion, the authors can add more description or combine those sentences with other paragraphes.

According to the suggestions from the reviewer, we combined the given three paragraphs and made one paragraph

  1. Fig 1 and Fig 2: Please increase resolutions of both pictures. Can the authors use real parameters to replace X1, X2, and Y1/Y3.

The brightest settings have been used in the statistics programme, we can send the original drawings both in the Statistics programme and in jipeg format. We have also added a fitted surface plot for the Y1 response - with factors X1 and X2, as well as factors X1 and X3, as they all affect the penetration level. The plots in the Statistics programme are called fitted surface, but the real experimental data are used to construct this plot.

Thank you again for your comments, we hope we have answered all your questions. Best regards, authors.

Reviewer 3 Report

I enjoyed reading the manuscript entitled “Development of a technology for protein-based, glueless Belevskaya pastille with study of the impact of probiotic sour-dough dosage and technological parameters on its rheological properties”. Manuscript is not too many organized and many proposed changes must be done in different parts of the manuscript. In my point of view, the document is perfect with very major comments/corrections that have to be modified.

My comments are as follow:

·         The abstract section needs to be improved, specially at the end, with more accurate information about the conclusion of your research. Moreover, the conclusion section of your manuscript is very similar that the abstract, therefore, please modify both of them.

·         In lines 36 and 37, in the keywords section. Since some words are repeated in the manuscript title and in the keyword section, I suggest to replace the words: “probiotic” or “rheological properties”, by “Hardness analysis”, “moisture content” or other keywords, to have more visibility of your manuscript paper.

·         In the Introduction section, please, improve it. It is to elementary and undeveloped.

·         In Table 1, please, include the coded values for each factor (independent variables).

·         In line 144, what version of Statistica software did you use?

·         In the Statistical Analysis section, did you the Levene's tests were performed for equality of variances and Shapiro-Wilk's tests to evaluate normal distribution. Nevertheless, what about the analysis of “autocorrelation in residuals”? Did you apply it? Why not?

·         Moreover, in the statistical analysis section, in lines from 168 to 175 you never explained about the “surface response methodology” (SRM) that you applied in the Results and discussion section, in tables 6, 7, 8, 9 and figures 1, 2 .

·         Please, improve the discussion section of all your results including more information related to: i) Determination of lactic acid bacteria in whipped protein pastille and study of their probiotic properties; ii) Influence of technological factors on the growth and development of lactic acid microorganisms by the method of mathematical modelling; iii) Influence of technological factors on the content of lactic acid organisms in whipped protein pastille.

·         Furthermore, if you applied a SRM to analyse your data, did you apply a validation of the SRM optimization? In the optimization analysis, what method did you apply to compare your experimental data against the optimal values?

·         The conclusion section looks like “another summary”, please modify it accordingly.

·         In general, the document is very disordered. Tables and Figures do not have a clear title information and them do not have a properly legend. The equations in lines 309, 315, 316, 318 and 319 are very messy. From line 384, the manuscript has a blank space.

Author Response

Thank you for your time and your recommendations for improving our article. Your comments are very valuable to us. After all, they will help us to make our article better.

We have prepared answers to your questions:

  1. The abstract section needs to be improved, specially at the end, with more accurate information about the conclusion of your research. Moreover, the conclusion section of your manuscript is very similar that the abstract, therefore, please modify both of them.

We have changed the abstract and added more conclusions about our study, hopefully now it is better. We have also made our conclusion more meaningful. In the corrected version, it was added the following paragraph:

“The proper functioning of the gastrointestinal tract plays an important role in strengthening the immune system. It is an undeniable fact that lactic acid microorganisms are necessary for the proper functioning of the gastrointestinal tract, the source of which are mainly dairy products. However, there is a problem with the digestibility of lactose; therefore, alternative sources and carriers of probiotics are of particular interest. Due to its dietary and natural properties, protein marshmallow can serve as such a carrier. Therefore, the direction of this study is to identify the dependence of technological factors on the rheological properties of the product and the growth of lactic acid microorganisms in confectionery products enriched with lyophilized strains.”

  1. In lines 36 and 37, in the keywords section. Since some words are repeated in the manuscript title and in the keyword section, I suggest to replace the words: “probiotic” or “rheological properties”, by “Hardness analysis”, “moisture content” or other keywords, to have more visibility of your manuscript paper.

Thank you for your comment. According to this comment, the words "probiotic" and "rheological properties" were replaced in the keywords section by "drying time", "moisture content".

  1. In the Introduction section, please, improve it. It is to elementary and undeveloped.

In the introduction, the structure was improved and more extensive information on the study with sources was added, consequently, the number of literature references was changed and the review part of the paper was increased. It was added the following paragraphs:

“It is also noted that in the original version of the recipe, there were no sweeteners; only lat-er, to increase the sweetness was honey added to the pastille mass, which was then re-placed with sugar [3].

The main raw material for the production of whipped protein pastilles is apple puree, which is obtained from baked apples, in which a special form of fiber is pectin. The action of pectin is more pronounced during the heat treatment of apples, and this process is accompanied by the acquisition of a gel-like sheen on the surface of apples. It is known that the use of applesauce as the main prescription component carries a number of positive aspects. Emerging evidence suggests that pectin may help repair and preserve the intestinal mucosa [4]. Pectin also helps in the modulation of intestinal bacteria, eliminates toxins in the intestines and reduces inflammation [5].

A study was conducted showing that the chemical and nutritional properties of apples from 12 varieties of apples manufactured industrially (heat treatment at 95 °C for 2 minutes, pasteurization at 90 °C for 5 minutes, and cooling at 20 °C for 20 minutes) are very close to those of fresh apple pulp [6].

The topic of probiotic products is an integral part of the nutrition industry. Eating confectionery products produced using lactic acid bacteria (LAB) contribute to positive changes in the intestinal microbiota.

The results of the authors' research [7] showed that, despite some loss of probiotic adhesion, the combined presence of extract and probiotic is more effective in reducing the overall amount of adhered viable pathogen cells than the probiotic alone, regardless of the probiotic/pathogen system considered.

A review article [8] shows that the potential of probiotics is to increase innate and ac-quired immunity and activate anti-inflammatory and anti-allergic effects. This study shows the relevance of the use of probiotics in human life. However, the question remains unresolved regarding how easy and affordable it is to consume them in everyday life, particularly for people suffering from the indigestibility of milk protein. In [9], the authors proved the possibility of producing probiotic grape marmalade without the use of dairy products.

Previous works [10] studied the influence of technological parameters in the production of marmalade products on the safety of vital nutrients. In their further works [24], the morphology of the probiotic culture was studied, and it was found that the volume of whey (250 ml) affects the growth of LAB in the marmalade product. The optimal amount of probiotic culture to be added to the serum was 0.01–0.02 g, while the time of reviving the microorganisms was 6 hours. As a result of the study, the confectionery product contained from 1 to 3 CFU/g lactic acid culture and 1.7 and 2.2 times more antioxidants com-pared to the control, which justifies the benefits for the gastrointestinal tract and maintaining human immunity due to the production of its own interferon.

The studies conducted in this area prove the promise of this direction; however, it is necessary before previous to industrial applicability, it is necessary to study technological methods for introducing a probiotic culture as an active enriching ingredient, followed by a study of the safety of LAB after technological processing and storage. It is also necessary to analyze exactly what effect the dosage and type of probiotic culture has on the rheological properties of finished products. [11].

To identify the dependence of technological factors on the rheological properties of the product and the growth of LAB in battered confectionery products, it was set 3 objectives:

  1. It is necessary to determine the optimal dosage of probiotic starter, which would allow enrichment of the product with probiotics and, at the same time, would not worsen the rheological and organoleptic properties of finished products;
  2. The number of surviving LAB in Belevskaya whipped protein pastilles after processing treatment should be investigated to determine the effectiveness of their probiotic properties and stability in acidic medium;
  3. The influence of technological factors on the rheological properties of Belevskaya whipped protein pastilles and the growth and development of LAB should be substantiated using mathematical modeling, revealing the significance of factors by means of dispersion and regression analysis.”
  4. In Table 1, please, include the coded values for each factor (independent variables).

Table 1 indicated the coded values for each factor (independent variables). X1; X2 and X3 were added.

  1. In line 144, what version of Statistica software did you use?

We used version 13 of Statistica software and specified on the line 239

  1. In the Statistical Analysis section, did you the Levene's tests were performed for equality of variances and Shapiro-Wilk's tests to evaluate normal distribution. Nevertheless, what about the analysis of “autocorrelation in residuals”? Did you apply it? Why not?

In the statistics programme you can quickly do an analysis of variance and determine the significance of the factors by the P value. The P value should be lower than 0.05. We also looked at the Pareto chart of standardised effects, which shows the significance of factors and their interaction.

Autocorrelation of residuals is only applied to time-series data. Our data are experimental and they are independent of time (time series)

  1. Moreover, in the statistical analysis section, in lines from 168 to 175 you never explained about the “surface response methodology” (SRM) that you applied in the Results and discussion section, in tables 6, 7, 8, 9 and figures 1, 2 .

Analysis of variance proves the relationship between the resultant and factor attributes.

The analysis of variance was used to determine the influence of technological factors such as drying time, temperature and amount of inoculum on the moisture content of finished products because it shows the deviation of experimental data from the mean values.

Regression analysis shows the effect of predictors X1, X2, X3 on the resultant attribute U2 and the identification of regression model (equation).

Regression analysis reveals the relationship of the resultant from the predictors (X1, X2, X3), as well as assesses the significance of the model (lines 243-248).

  1. Please, improve the discussion section of all your results including more information related to: i) Determination of lactic acid bacteria in whipped protein pastille and study of their probiotic properties; ii) Influence of technological factors on the growth and development of lactic acid microorganisms by the method of mathematical modelling; iii) Influence of technological factors on the content of lactic acid organisms in whipped protein pastille.

We've made an addition to the discussion section. In concrete, we have added the following paragraphs:

“The results of previous works [28,29] confirm that LAB play an important role in the inhibition of food pathogenic and putrefactive microorganisms due to antimicrobial metabolites.”

“Meanwhile, a study by the authors [34] showed that Lactobacillus acidophilus is tolerant to pepsin, an enzyme of gastric juice.”

“As seen from Figure 1, the higher the temperature and longer the drying process, the lower the penetration level becomes. The volume and texture of the products become denser. Figure 2 shows that the higher the dosage of probiotic inoculum and the shorter the drying time, the higher the penetration level becomes and the more crumbly the product becomes. Therefore, more than 0.021% of probiotic starter should not be added to protein pastilla, as the rheological and organoleptic properties of finished products will deteriorate.”

“Regression analysis shows the influence of predictors X1, X2, and X3 on the resultant attribute Y2 and the identification of the regression model (equations).

Regression equation: y2=110,41 ‒ 4.74x1 ‒ 1.13x2

y2=110,41 ‒ 4.74× 12  ‒ 1.13× 50=-2,97 for the minimum value

y2=110,41 ‒ 4.74× 16  ‒ 1.13× 70=-44,53 for the maximum value

The regression equation clearly demonstrates that the more processing takes place, the lower the moisture content of the finished products will be, despite the presence of any amount of probiotics in its composition.”

“Table 9 shows that the only factor affecting the mold is the dosage of probiotic starter applied. However, this study was conducted with a control sample with no inoculum. If we analyze without a control sample, only between the minimum and maximum dosage, then there will be another significant factor affecting the survival of probiotics, and this factor is temperature.”

“The advantage of the study is that we proved the survival of probiotic cultures in the developed whipped protein pastille, which can provide a carrier of probiotics as an alter-native to dairy products. The optimal dosage of the probiotic was determined, which al-lowed enrichment of the product and maintenance of the proper rheological and organo-leptic properties necessary for industrial production.”

  1. Furthermore, if you applied a SRM to analyse your data, did you apply a validation of the SRM optimization? In the optimization analysis, what method did you apply to compare your experimental data against the optimal values?

We did not use the model with the construction of task optimisation and removed this expression from the paper

  1. The conclusion section looks like “another summary”, please modify it accordingly.

We have changed the conclusion section, hopefully making it clearer. The new version of conclusions reads as follows:

“In this study, it was found that bile protein marshmallow can serve as a carrier for probiotic enrichment and be an alternative to dairy products, as it was found that micro-organisms of the genus L. acidophilus survived after processing treatment and that L. acidophilus M3 and L. acidophilus M4 strains were the most resistant to high bile concentrations in the substrate. The optimal technological mode of preparation of protein pastille (drying the pastille mass in a dehydrator at a temperature of 50 °C for 16 hours.) and the optimal dosage of probiotic (0.021%), which allowed enrichment of the product with lactic acid bacteria and preserved the organoleptic and rheological properties of the finished products, were revealed. Based on the analysis of variance and the regression equations obtained, it was found that the growth of LAB in the product was strongly influenced by the amount of sourdough starter applied (coefficient of determination R² = 0.96). The penetration level was influenced by factors such as the amount of introduced probiotic starter, drying time and the interaction between the factors of drying time and the amount of introduced starter.

In general, the results of the study showed the effect of probiotic cultures on the rheo-logical properties of protein pastilles, more than the established dosage should not be ap-plied, as the higher the dosage, the more crumbly and friable the product becomes, which will optimize the processes of industrial production. This study provides an opportunity to use the obtained data on a larger scale and is applicable for other products similar in technological process.”

  1. In general, the document is very disordered. Tables and Figures do not have a clear title information and them do not have a properly legend. The equations in lines 309, 315, 316, 318 and 319 are very messy. From line 384, the manuscript has a blank space.

We have tried to organize the article, tables and figures.  In the equations, according to Table 6, the combination of factors X₁X₂ (duration and drying temperature), X₂X₃ (drying temperature and the amount of inoculum) has the least influence, so we removed from the formula the factors that have no influence on the penetration level.

Thank you again for your comments, we hope we have answered all your questions. Best regards, authors.

Round 2

Reviewer 2 Report

This manuscript improved.

Reviewer 3 Report

I enjoyed reading again the manuscript entitled “Development of a technology for protein-based, glueless Belevskaya pastille with study of the impact of probiotic sour-dough dosage and technological parameters on its rheological properties”. In this second round for the review of the above-mentioned manuscript, the document is well organized and most of the suggestions indicated in the first-round manuscript correction were amended. In my point of view, the manuscript is accepted in present form.